# Long-Term Outcomes of Staged Revision Surgery for Chronic Periprosthetic Joint Infection of Total Hip Arthroplasty

**DOI:** 10.3390/jcm11010122

**Published:** 2021-12-27

**Authors:** Christopher W. Day, Kerry Costi, Susan Pannach, Gerald J. Atkins, Jochen G. Hofstaetter, Stuart A. Callary, Renjy Nelson, Donald W. Howie, Lucian B. Solomon

**Affiliations:** 1Department of Orthopaedics and Trauma, Royal Adelaide Hospital, Adelaide, SA 5000, Australia; daychrisw@gmail.com (C.W.D.); susan.pannach@sa.gov.au (S.P.); stuart.callary@sa.gov.au (S.A.C.); donald.howie@sa.gov.au (D.W.H.); bogdan.solomon@sa.gov.au (L.B.S.); 2Centre for Orthopaedic and Trauma Research, Faculty of Health and Medical Sciences, The University of Adelaide, Adelaide, SA 5005, Australia; gerald.atkins@adelaide.edu.au; 3Ludwig Boltzmann Institute of Osteology, Hanusch Hospital of OEGK and AUVA Trauma Centre Meidling, 1st Medical Department Hanusch Hospital, 1140 Vienna, Austria; jochen.hofstaetter@gmail.com; 4Michael Ogon Laboratory, Orthopaedic Hospital Vienna-Speising, 1130 Vienna, Austria; 5Department of Infectious Diseases, Royal Adelaide Hospital, Adelaide, SA 5000, Australia; Renjy.Nelson@sa.gov.au

**Keywords:** total hip arthroplasty, periprosthetic joint infection (PJI), eradication, two-stage revision, functional outcomes, mortality

## Abstract

Periprosthetic joint infection (PJI) is a serious complication of total hip arthroplasty. Staged revision surgery is considered effective in eradicating PJI. We aimed to determine the rate of infection resolution after each stage of staged revision surgery (first stage, repeat first stage, second stage, excision arthroplasty, and reimplantation) and to assess functional outcomes and the mortality rate at ten years in a consecutive series of 30 chronic PJI of total hip arthroplasties. Infection resolution was defined as no clinical nor laboratory evidence of infection at 24 months after the last surgery and after a minimum of 12 months following cessation of antimicrobial treatment. Four patients died within 24 months of their final surgery. Nineteen patients, 73% (worst-case analysis (wca) 63%), were infection free after 1 surgery; 22 patients, 85% (wca 73%), were infection free after 2 surgeries; and 26 patients, 100% (wca 87%), were infection free after three and four surgeries. The median Harris Hip Score was 41 prior to first revision surgery and improved to 74 at twelve months and 76 at ten years after the final surgery. Thirteen patients died at a mean of 64 months from first revision, giving a mortality rate of 43% at ten years, which is approximately 25% higher than that of an age-matched general population. The results show that with repeated aggressive surgical treatment, most PJIs of the hip are curable. Ten years after successful treatment of PJI, functional outcomes and pain are improved and maintained compared to before initial surgery, but this must be balanced against the high 10-year mortality. Level of evidence: cohort studies.

## 1. Introduction

Periprosthetic joint infection (PJI) is a serious complication following joint arthroplasty surgery. Importantly, PJI is one of the few complications of joint arthroplasty with an increasing incidence [1,2,3]. In addition, management of PJI remains controversial [4], failure rates continue to be high [5,6], outcomes of treatment were shown not to have improved over time [7] and the long-term joint function and quality of life of these patients is low [8]. The collaboration between surgeons and infectious diseases and microbiology consultants in the diagnosis and treatment of PJI is seen as essential for improvements in the surgical and medical management of this condition [9]. The best diagnostic criteria and management options for PJI remain controversial [10,11,12]. PJI is currently diagnosed by a set of criteria agreed to at the Second International Consensus Meeting on Musculoskeletal Infection [12]. Major, definitive criteria include either the presence of two positive cultures of the same organism or the presence of a sinus tract communicating with the joint or prosthesis. Diagnosis can also be made on a cumulative score of ≥6 points from several minor criteria: elevated serum CRP or D-Dimer (2 points) and ESR (1 point); elevated synovial WBC or LE (1 point); PMN (%) (2 points); CRP (1 point); positive alpha-defensin (3 points); positive histology (3 points); purulence (3 points); single positive culture (2 points). Regarding management, controversies remain as to the length, type and route of administration of antibiotic, as well as surgical treatment [10].

PJI is a major cause of revision following total hip arthroplasty (THA), accounting for 10–15% of all revisions [1,13,14] and is commonly treated surgically by debridement antibiotics and implant retention (DAIR), single-stage revision or multi-staged revision [14,15]. Multi-staged revision involves a first stage, where, after implant removal and debridement, the patient is managed either with an excision arthroplasty or implantation of an antibiotic-coated interval prosthesis for a period until the patient is deemed infection free. Compared with resection arthroplasty or implantation of a static spacer, the articulating design of the interval prosthesis is favored since it maintains limb length and soft tissue tension and facilitates patient’s ambulation between revision surgeries [15,16,17]. Furthermore, meta-analysis has shown eradication rates with two-stage revision to be consistently high at over 90%, giving the best chance of cure over other treatment options [16]. However, patients undergoing multi-staged revision are not always limited to having two surgeries. When the patients are deemed not to be infection free after the first-stage revision, repeat ‘first stages’ are performed, where new interval prostheses are implanted in an effort to resolve the infection before a final revision surgery where a definitive prosthesis is implanted. Alternatively, excision arthroplasty is performed following a failed first-stage revision. Although a two-stage revision with an antibiotic-coated interval prosthesis at the first stage is currently regarded as the gold standard of treatment in PJI, some studies report equal cure rates after DAIR and single-stage revisions [10]. In addition, it is recognized that multiple major surgeries can have a negative effect on the functional and quality of life outcomes of these patients [8].

Patients who suffer from PJI of a hip arthroplasty are at high risk of poor function, increased morbidity and death [8,18,19]. Overall, little is known about the long-term outcomes of patients who suffer PJI [8,10,11]. What is known, however, is that infection resolution decreases over time, with recurrence rates of 15% at 15 years [20]. Moreover, even poorer maintenance of infection resolution of 47% at ten years in polymicrobial infections [21] and 65% at five years in culture-negative infections [22] has been reported. Given this, more studies of longer-term outcomes will help improve disease management strategies. This is important not only for improving patient outcomes but also to combat the rising health care costs to treat PJI [23]. The present study aimed to: (1) determine infection resolution rates after each stage of a multi-staged revision for PJI of the hip; (2) assess functional outcomes at ten years; and (3) determine the ten-year mortality rate for a PJI in a hip arthroplasty cohort.

## 2. Materials and Methods

### 2.1. Study Participants

This single-centre cohort study at a tertiary-referral public hospital was approved by the ethics review board of the institution (Approval No. 010310a). In this retrospective analysis of prospectively collected data, 30 consecutive infected hip arthroplasty patients, consisting of 17 primary total hip arthroplasties (THAs), 10 revision THAs and three hemiarthroplasties (HAs), consented to undergo staged revision surgery between 2005 and 2011. According to the staging system described by McPherson et al. [24], all patients presented with a late-chronic infection (>4 weeks, type III). When retrospectively applied, most patients met MSIS criteria for PJI [12], except for two who met only three minor criteria. The cohort comprised 16 males and 14 females with a mean age of 67 years (range 43–84 years) and a mean body mass index (BMI) of 31 (range 18–51). Patients’ ASA physical status [25] and Charlson Comorbidity Index (CCI) [26,27] are summarized in Table 1. Fourteen patients had severe systemic disease and 16 patients had a moderate to severe CCI score. 

The start date for this study was chosen as the time we started using commercially available hip prostheses as an interval prosthesis, as opposed to routine excision arthroplasty or an artisanal interval spacer at the first stage. The end date was determined to allow for a minimum of 10-year follow up. The mean time between the primary/revision arthroplasty and first-stage revision was 52 months (range 1–177 months). Sixteen of the patients were initially treated with a DAIR, which failed, before undergoing staged revision surgery. The remaining 14 patients had no prior surgery to treat the PJI, besides the ones described below after being diagnosed with a PJI. Each revision surgery was aimed to be curative of infection. Surgery was performed by, or under the supervision of, any one of four orthopaedic surgeons specialising in hip reconstruction.

### 2.2. First Surgery—First-Stage Revision

At first-stage revision, treatment consisted of removal of all foreign material, thorough debridement and irrigation, and implantation of an antibiotic-coated interval THA. Details on the surgical technique have been described previously [28,29,30]. The temporary THA was an Elite-Plus^®^ antibiotic cement-coated stem (DePuy, Warsaw, IN, USA) with a cemented polyethylene liner in the first six hips and then a PROSTALAC^®^ implant system (DePuy, Warsaw, IN, USA) in the next 24 cases. These implants allowed for a mould-based temporary prosthesis, thereby improving stability by three-point fixation and proximal cementation compared to older techniques of temporary spacer implantation. The implants were surrounded by antibiotic-loaded cement with the addition of 3 g of vancomycin per 40 g bag of antibiotic bone cement. Commercially available Simplex P (Stryker, Mahwah, NJ, USA) cement containing 1 g of tobramycin was used in the first 23 hips and Palacos^®^ R+G (Heraeus, Wehrheim, Germany) cement containing 0.5 g of gentamicin in the last seven. The cement was hand mixed in each case. A minimum of five periprosthetic tissue samples for microbiological culture and additional samples for frozen section and routine histology were taken intraoperatively in all patients. Patients were monitored at various time points after first-stage surgery with clinical examination, radiographs, and measurement of CRP levels.

### 2.3. Second Surgery—Repeat First-Stage/Second-Stage Revision

In order to proceed to second-stage revision, a patient had to be free of infection as guided by the clinical picture and laboratory findings, otherwise a repeat first-stage surgery was performed. Second-stage revision consisted of removal of the temporary implant, debridement and irrigation, and implantation of the definitive prosthesis. Simplex P (Stryker, Mahwah, NJ, USA) antibiotic bone cement with addition of 0.5 g vancomycin/40 g bag of cement (containing 1 g of tobramycin) was used if a cemented implant was inserted. Tissue samples were routinely taken for microbiology, frozen sections, and routine histopathological examination. A repeat first-stage surgery with another interval prosthesis was performed in all cases deemed to still be infected, except in the case of a mycotic infection, when an excision arthroplasty was performed. 

### 2.4. Third and Subsequent Surgery—Repeat First-Stage/Second-Stage Surgery

For patients for whom the first- or second-stage procedure failed, the above protocol was repeated. Patients thought to have a possible persistent infection following a repeat first-stage revision underwent a resection arthroplasty. 

### 2.5. Antibiotic Treatment

Antibiotic treatment was prescribed by an infectious diseases specialist based on culture results. The interval between planned staged surgeries was 3 months, except for mycotic infections. Antibiotic treatment included 6 weeks of intravenous antibiotic followed by 6 weeks of oral antibiotic. Antibiotics were continued until the second surgery excluding the two patients deemed unfit to undergo the second-stage revision. For these patients the oral antibiotics were stopped at 3 months after the first-stage revision. For the mycotic infection, antibiotic treatment and reimplantation surgery were extended to 2 years. For patients undergoing a second-stage revision, intravenous antibiotics were continued post-operatively until definitive culture results were available 14 days after surgery.

### 2.6. Determination of Infection Cure and Patient Follow Up

Patients were classified as infection free if they returned less than 2 positive cultures with the same organism out of 5 samples with a low virulence organism, and 0 positive cultures out of 5 samples with a high virulence organism [12,31] at last revision surgery and, at a minimum of 24 months after last revision surgery, there had been no subsequent re-operation for infection, no antibiotic treatment for a minimum of 12 months, no clinical signs of infection, and normal CRP levels. Patients who were on continuing antibiotic treatment or were <12 months since cessation of antibiotic treatment but had no clinical or laboratory signs of infection were classified as ‘probably infection free’.

Radiographs, pathology results, and clinical scores were obtained on patients pre-operatively and post-operatively at scheduled follow up after each procedure. The Harris Hip Scores (HHS) and Pain Scores [32,33] and Société Internationale de Chirurgie Orthopédique et de Traumatologie (SICOT) Activity Scores [34] were obtained to determine functional outcomes up to ten years following treatment. A Wilcoxon matched-pairs signed rank test was used to determine improvement in post-operative scores from pre-operative second-stage baseline values. *p* values < 0.05 were considered statistically significant. Mortality rate for the original 30 patient group was determined by deaths occurring within ten years of the most recent revision surgery, regardless of the cause.

## 3. Results

No patients were lost to follow up. Four patients died less than 24 months after their last surgery. The surgical and infection pathways of the 30 patients are summarized in Figure 1.

Patients underwent up to four surgeries (Table 2).

At the first surgery/first-stage revision, a single microorganism was identified in 22 of 30 cases (73%) and multiple microorganisms were identified in 6 of 30 cases (20%). In only two patients could no microorganism be identified. Gram-negative species were detected in six patients (20%) (Table 3). Most common were coagulase-negative staphylococci (CNS) (27%) and/or *S. aureus* (27%), identified in 19 (63%) of the infected hips. Methicillin-resistant staphylococcal strains (MRSA) accounted for three (10%) of these infections. 

Twenty four of the 30 patients were thought to be probably infection free after their first surgery/first-stage revision. Two of these patients died before a second-stage revision surgery, with <24 months follow up and were excluded from the outcomes analysis. Of the remaining 22 patients, 20 proceeded to second-stage revision at a minimum of 3 months (mean of 5 months, range 3–22) after their first-stage operation. Time to second-stage revision was often delayed due to incorporating management of these cases around elective and semi-elective surgery waitlists. The remaining two patients were medically unfit to undergo second-stage surgery but remained infection free as per clinical and laboratory findings for >24 months with the interval prosthesis remaining in situ. Of the 20 patients that proceeded to a second-stage revision as the second surgery, two patients died with <24 months follow up and were excluded from the outcomes analysis. A third patient had a persistently elevated CRP and reported increasing pain at one year after the second-stage revision. The patient subsequently underwent re-operation, at which time intraoperative cultures were found to be positive with a different organism (Case 8, Table 4). It is unclear if this patient was either never cured of an originally undiagnosed polymicrobial infection, was reinfected at the time of the second-stage revision or suffered a late re-infection after second-stage revision. The patient was successfully treated with a subsequent repeat two-stage revision (Figure 1).

Six of the 30 patients were thought to have a possible persistent infection after the first surgery, a first-stage revision. All six were patients that underwent a DAIR before undergoing revision surgery. Four of these patients had a PJI of a primary THA, while the other two had a PJI of a revision THA. The associated microorganisms found in these cases are listed in Table 3. One patient whose cultures initially grew *P. aeruginosa* had a subsequent hip aspirate from which *Candida albicans* was grown. This patient underwent a resection arthroplasty without undergoing a repeat first-stage revision. Twenty-four months later, and 3 weeks after suspension of antibiotic treatment, a repeat deep biopsy was negative for any microorganism. Twelve months after this biopsy, the patient had no clinical or laboratory sign of infection, underwent revision THA and remained infection free at 60 months follow up. The other five patients had an ongoing elevated CRP and/or >5 PML/high-power field on frozen sections at reoperation and underwent repeat first-stage revision at a mean of 14 weeks (range 10–25 weeks). Of these, two patients were thought to be infection free and proceeded to second-stage revision, whereas three patients were deemed to have a possible persistent infection after the repeat first-stage revision and underwent resection arthroplasty. Two of the three patients who had a resection arthroplasty subsequently underwent revision THA, and the third patient underwent treatment for a thoracic malignancy and died infection free before undergoing re-implantation of a THA.

One of the 24 patients considered infection free after their first surgery, a first-stage revision, who went on to have a successful second-stage revision as the second surgery, continued to have no clinical and laboratory symptoms and signs of infection until 45 months after the second surgery when he developed an *E. coli* bacteremia secondary to an acute urinary tract infection. This was followed by a haematogenous reinfection of the revised hip (Case 7, Table 4). The patient was medically unfit to undergo further revision surgery and, after DAIR, was placed on lifetime suppressive antibiotics.

In only one case was the same microorganism, an enterococcus, identified during the second procedure. In 20 of the 26 hips that underwent more than one revision, all cultures at the last surgery were negative. Furthermore, only one of the five cultures from each of the six possible persistent infected hips was positive and were considered contaminations. The overall infection resolution, and that after each stage, is reported in Table 5 as best-case analysis (BCA) and worst-case analysis (WCA). Resolution of infection was achieved in 73 to 77% (WCA 63 to 67%) of infected hip arthroplasties after the first surgery, first-stage revision, in 85 to 88% (WCA 73 to 77%) after the second surgery and 100% (WCA 87%) after the third surgery.

Twenty six of the 30 patients had a minimum follow up of 24 months (mean 104, range 28–171 months) after their last revision surgery. Of these, 25 (96%) had a functioning THA. The median pre-operative HHS score was poor prior to the first revision (41, range 13–83) and second revision (56, range 19–92) (Figure 2). For the 23 patients given a final definitive THA at the last revision, there was a fair improvement in the median HHS at 12 months (74, range 12–97), 24 months (78, range 23–97) and at 120 months (76, range 48–88). After the last revision, there was a significant improvement in function at 12 months (*p* = 0.004) and 24 months (*p* = 0.010) relative to that recorded pre-operatively.

The median post-operative pain score improved to 40 (range 0–44) at 12 months, 44 (range 10–44) at 24 months and 44 (range 30–44) at 120 months following last revision (Figure 3). After the last revision, there was a significant improvement in pain at 12 months relative to that recorded pre-operatively (*p* = 0.046).

Median patient-reported activity after last revision increased by one rating, from sedentary to semi-sedentary between 3 and 36 months and up to light labor at 48, 72, and 84 months post-operatively (Figure 4). After the last revision, there was a significant improvement in activity at 12 months relative to that recorded pre-operatively (*p* = 0.042).

Similar functional outcomes were reported in patients undergoing either two-stage or multi-staged revisions, with slightly better outcomes of pain and activity in patients having two-stage revisions (Table 6). For the five patients who underwent multi-staged revisions there were improvements in HHS, Pain and activity at 12 months, 24 months and at latest follow up compared to pre-operatively after the first-stage revision. 

Overall, 13 of the original 30 patients died at a mean of 64 months (median 74, range 1–139) from the first-stage revision, giving a mortality rate of 43.3%. The causes of death are listed in Table 7. Cause of death was unobtainable for four patients. However, all four patients had multiple comorbidities (CCI ≥ 4) including acute renal failure, cardiac disease and cancer at their last known admission prior to death. Ten of the 13 cases had moderate (3–4) to severe (≥5) CCI scores.

Two of the three (66%) patients with a PJI after a HA, seven of 17 patients (41%) with a PJI after a primary THA, and 4 out of 10 patients (40%) with a PJI after a revision THA died. Both patients that died early, before undergoing the second-stage surgery, had a PJI after a HA. The third patient with a PJI after a HA was one of the patients deemed unfit to undergo a second surgery, but who lived more than 24 months after the first surgery, a first-stage revision.

## 4. Discussion

The goals of our study were threefold. First, we aimed to determine the infection resolution rates after each stage of staged revision surgery, and whether this was maintained over time. Next, we assessed functional outcomes, with long-term HHS and SICOT activity scores up to ten years after treatment. Finally, we set out to determine the long-term mortality rate for this cohort.

Resolution of infection was achieved in 73 to 77% (WCA 63 to 67%) of infected hip arthroplasties after the first surgery, first-stage revision, in 85 to 88% (WCA 73 to 77%) after the second surgery and 100% (WCA 87%) after the third surgery. The literature reports the best results of infection resolution after two-stage revision THA between 89% and 96% [28,30,35,36,37]. Our results include those for infections with antibiotic-resistant pathogens, MRSA and MSRE, for which a lower rate of infection resolution has been reported [38]. Further, the incidence of failure after first-stage revision surgery is often difficult to ascertain from the literature because it is not clear in some studies whether these were simply not reported or did not occur [28,30,36,37]. 

In only one of the hips with persistent infection after the first surgery was the same microorganism identified. Hence, if biological infection eradication of the original organism(s) was considered as the single measure of success, infection resolution after the first-stage revision increased to 97%. This figure should, however, be treated with caution, as the literature reports that up to 28% of PJI after THA are culture negative [39]. The high rate of success here could be explained in two ways. The initially infecting microorganism could have been eradicated by the first-stage revision and the persistent infection was a new infection caused by different organism(s) or, alternatively, the persistent infection could have been due to a microorganism not identified at the first-stage revision. All six patients that failed to be infection free after their first-stage revision had undergone a previous DAIR procedure, which may have suppressed the infecting microorganism, for example, by promoting small colony variant formation and/or an intracellular infection refractory to culture [40]. 

Interestingly, one of our patients suffered a subsequent PJI with a different pathogen one year after apparent cure after a two-stage revision. We cannot be certain if this patient was either never cured of his original infection, was reinfected at the time of the second-stage revision or suffered a late reinfection after the second-stage revision. The patient was successfully treated with a repeat two-stage revision. In this series, in seven out of eight cases diagnosed infected after the first surgery, first-stage revision, a new or additional organism(s) was cultured, suggesting either that the original diagnoses were incomplete due to an initially complex polymicrobial infection or that this patient sub-group was hyper-susceptible to PJI, perhaps due to immune insufficiency. 

Four patients who were excluded from this study died within 24 months of their last revision surgery, thereby precluding a classification of being infection free. While none of their causes of death can be directly linked to their PJI, it is possible these patients had an underlying comorbidity because of, or as a contributing factor to, their PJI. 

The reasons for the success rate of infection resolution in our cohort are likely to be multifactorial. Aggressive surgical debridement at every surgery is of utmost importance, as is reflected in the significantly poorer outcomes in the patients who underwent a DAIR before revision surgery. In addition, not all patients achieved infection resolution after the first revision, highlighting the difficulty of disease management and diagnosis in cases with chronic infections. Several variables might have, however, influenced our results. Sixteen patients had a DAIR and were commenced on antibiotics prior to having a staged revision. Importantly in our study, all failed first-stage revisions were in cases that underwent a prior DAIR. Failed DAIR is thought by some to influence the results of staged revisions for PJI [41,42]. The length of antibiotic treatment is also known to influence outcomes in PJI and the current trend is to reduce the length of intravenous treatment [10,43].

While the HHS scores in these patients ranked poor to fair, there was an improvement in scores from before first revision to after last surgery which was sustained up to ten years after surgery. A significant improvement was seen particularly by 12 and 24 months post-operatively. Similar improvements in HHS scores have been reported by other studies investigating management of PJI of THA by staged revision surgery at last follow up [44,45]. Although patients’ function did not improve greatly, the pain component of the HHS did improve with treatment, to no or minimal pain, and lasted out to ten years. SICOT activity scores also increased from before first revision to after last revision, showing an improvement in patient self-reported activity. Given the necessity of treatment of PJI, these findings are encouraging in the face of a difficult clinical scenario. Knowing that function after revision THA is poorer than after primary THA [46], we are also encouraged by the sustained improvement in functional outcome in our series, despite the high level of morbidity associated with PJI as a cause for revision and our aggressive surgical treatment of these cases.

Mortality rates after revision hip arthroplasty for infection are known to be high and despite this, they are thought to be underreported or de-emphasised [19]. Berend et al. [19] discussed how patients who die during the study period or before final follow up are often excluded from analysis, seemingly lessening the mortality rate of patients undergoing treatment of infected hip arthroplasty. Published mortality rates of various descriptions are reported in the literature as 4% after first-stage debridement [19], as a hazard ratio of 1.42 compared to non-infected hip revision patients [47], as 40% at five years in patients over 80 years old [48], as 19% at four years after second-stage revision in difficult-to-treat organisms [38], or as 48% during a study period of 13 years [19], to name a few. Recent work by Natsuhara et al. [18] showed a 21% five-year mortality rate for patients with an average age of 65 undergoing two-stage revision for infected THA. The mortality rate of 43% at ten-year follow up in our series, doubling from known five-year mortality rates, once again highlights the seriousness of PJI. Remarkably, these mortality rates match those reported recently in another cohort in Sweden, 45% at 10 years [8]. Additionally, these survival rates are comparable to those of patients of similar age for all cancers combined in Australia [49] and is 10% higher than an accurately predicted 10-year mortality index for the general population of the same age [50]. Although the numbers are small, it is worthwhile noting the difference in mortality rates after PJI for HA compared with those after primary and revision THA. This is not surprising considering the high mortality of patients suffering a fractured neck of femur [51] and in line with the literature comparing the outcomes of PJI after HA with that after THA [52]. Importantly, the high rate of infection resolution after aggressive surgical treatment and the functional improvement after these surgeries need to be considered and balanced against the high mortality in these patients. Both the comorbidities of these patients as well as the treatments for PJI could have contributed to the high mortality rate observed. Future, larger studies should investigate the independent role of these variables in the mortality of patients with PJI. 

The strengths of our study include the prospectively collected data with no loss to follow up, surgical consistency, and the fact that we were able to culture a pathogen(s) in all cases of PJI except one. The homogeneity of the data might help potential confounding factors. However, this study has several limitations. First, the sample size of this cohort of patients is small, which is not unusual in a single-centre study with a long-term follow up. Future multi-centre studies will help improve this limitation. There was also variability in treatment course that includes length of antibiotic therapy, the use of different bone cements and implants, and the number of surgeries required. Additionally, the timing between the primary and revision arthroplasty and then the first- and second-stage revisions could influence the outcomes as can the fact that more than half of the patients had a DAIR before undergoing staged revision. In addition, as many patients did not survive until ten years after treatment, reported functional outcome data are provided on fewer patients than entered this study. 

## 5. Conclusions

In conclusion, our results show an excellent infection eradication rate after staged revision surgery for PJI, and that eradication is maintained in the long term. While most THA PJIs are curable with this aggressive surgical treatment, this must be balanced with the associated high rate of morbidity and mortality. At ten years after treatment of PJI with staged revision, functional outcomes and pain are improved and maintained compared to pre-operatively. Despite effective treatment for PJI after hip arthroplasty, the ten-year mortality rate is very high and needs to be considered when recommending treatment options to patients.

## Figures and Tables

**Figure 1 jcm-11-00122-f001:**
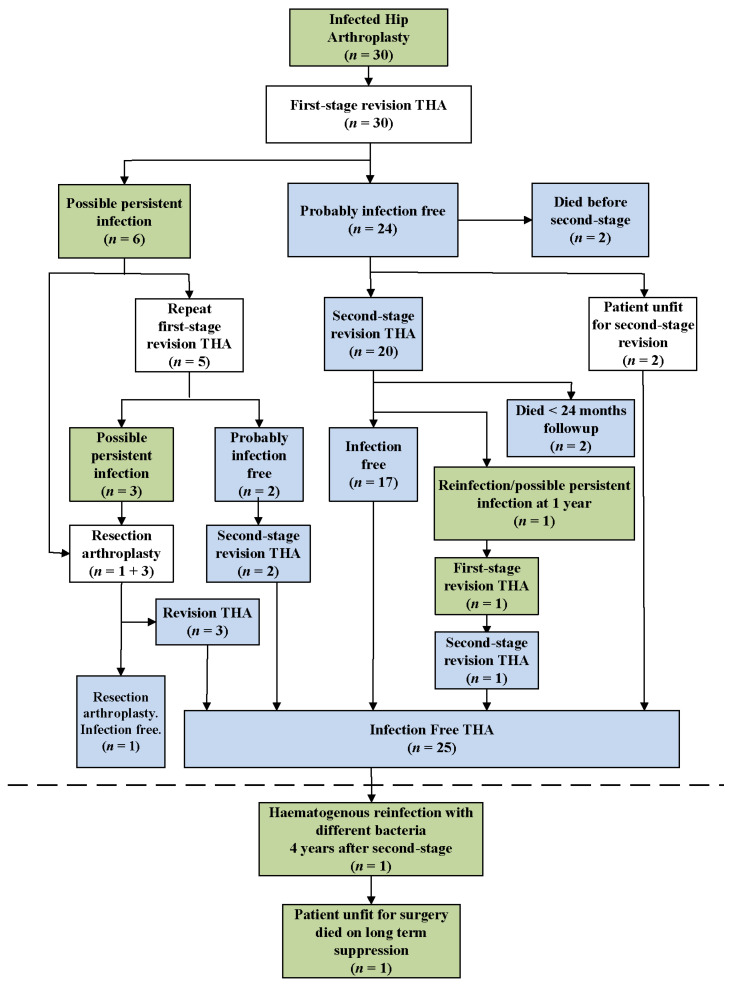
Surgical and infection pathways for study patients. Blue indicates an infection-free stage and green indicates a possibly infected or confirmed infection stage. Dashed line indicates end of treatment of the initial infection.

**Figure 2 jcm-11-00122-f002:**
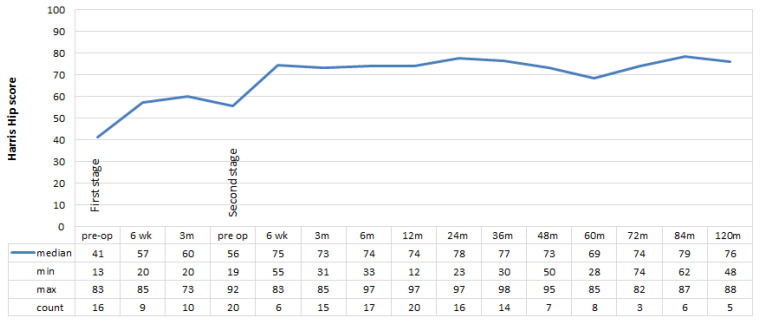
Harris Hip Scores. HHS: poor < 70, fair 70–79, good 80–89, and excellent 90–100.

**Figure 3 jcm-11-00122-f003:**
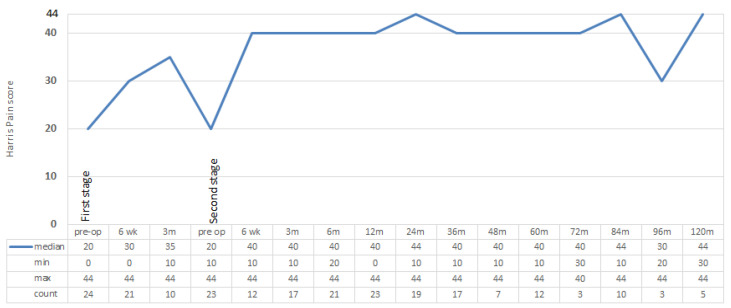
Harris Pain Scores. Harris pain: 44 = none, 40 = slight, 30 = mild, 20 = moderate, 10 = marked, and 0 = totally disabled.

**Figure 4 jcm-11-00122-f004:**
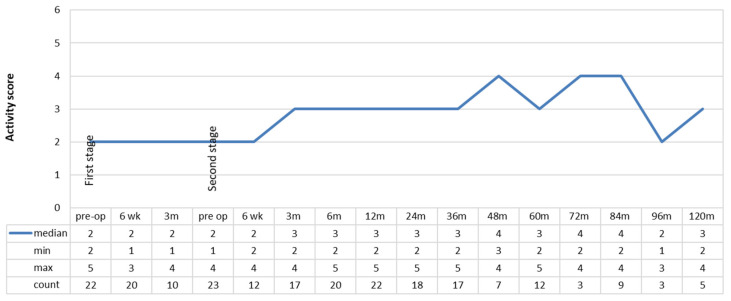
Activity level ratings using the SICOT activity score. SICOT activity: 1 = bedridden/wheelchair, 2 = sedentary, 3 = semi-sedentary, 4 = light labour, 5 = moderate manual labour, and 6 = heavy manual labour.

**Table 1 jcm-11-00122-t001:** Patient ASA and Charlson Comorbidity Index at first-stage revision.

	Number
ASA	
1 (normal healthy)	1
2 (mild systemic disease)	15
3 (severe systemic disease)	13
4 (severe systemic disease that is a constant threat to life)	1
Charlson Comorbidity Index	
0 (None)	6
1–2 (mild)	8
3–4 (moderate)	8
>5 (severe)	8

**Table 2 jcm-11-00122-t002:** Number and types of surgery in the study group.

	Type of Surgery	No. of Patients
First surgery	1st-stage revision	30
Second surgery	2nd-stage revision	20
	Repeat 1st stage	5
	Excision arthroplasty	1
Third surgery	1st-stage revision (after failed 2 stage)	1
	Excision arthroplasty	3
	2nd-stage revision	2
	Reimplantation after excision arthroplasty	1
Fourth surgery	2nd-stage revision	1
	Reimplantation after excision arthroplasty	2

**Table 3 jcm-11-00122-t003:** Microorganisms causing infection prior to first-stage revision.

Organism	Number of Hips *
CNS	8
MSSA	8
MRSA	3
*Enterococcus faecalis*	4
*Pseudomonas aeruginosa*	4
*Streptococcus viridans*	2
Group B Streptococcus	1
*Escherichia coli*	3
Enterobacter species	1
Mixed anaerobes	1

* Multiple organisms were identified in six hips, and therefore the sum of the different organisms exceeds the number of hips treated; CNS, coagulase-negative staphylococci; MRSA, methicillin-resistant Staphylococcus aureus.

**Table 4 jcm-11-00122-t004:** Microorganisms identified in the cases with possible persistent infection after first surgery, first-stage revision, or reinfection after multi-staged revision.

Case	Microorganisms at First Stage	Procedure	Microorganisms
1	*P. aeruginosa*	Resection arthroplasty after 1st stage	Candida albicans
2	*Enterococcus faecalis*	Repeat 1st stage	*Enterococcus faecalis, P. aeruginosa, Morganella morganii*
3	*E. coli*, CNS, *Enterococcus*	Repeat 1st stage	*Enterobacter cloacae*
4	*E. coli*, MSSA	Repeat 1st stage	CNS, *Candida parapsilosis*
5	CNS	Repeat 1st stage	No growth
6	MRSA	Repeat 1st stage	*P. aeruginosa*, *Klebsiella* sp.
7	*CNS, Cutibacterium acnes*	Debridement after 2nd stage	*E. coli*
8	MSSA	Repeat two-stage revision	CNS

CNS, coagulase-negative staphylococci; MSSA, methicillin-sensitive *Staphylococcus aureus*; MRSA, methicillin-resistant *S. aureus*; *P. aeruginosa*, *Pseudomonas aeruginosa*; MRSE, methicillin-resistant *Staphylococcus epidermidis*; *E. coli*, *Escherichia coli*; sp., species.

**Table 5 jcm-11-00122-t005:** Resolution of original infection.

Surgery/Stage	Probably Infection Free	Infection Free	
		BCA	WCA
1st surgery (1st-stage revision)	23/24 * of 30 (77–80%)	19/20 * of 26 (73–77%)	19/20 * of 30 (63–67%)
2nd surgery (repeat 1st stage/excision arthroplasty/2nd stage)	26/27 * of 30 (87–90%)	22/23 * of 26 (85–88%)	22/23 * of 30 (73–77%)
3rd surgery (repeat 1st stage/excision arthroplasty/reimplantation/2nd stage)	30 of 30 (100%)	26 of 26 (100%)	26 of 30 (87%)
4th surgery (reimplantation/2nd stage)	30 of 30 (100%)	26 of 26 (100%)	26 of 30 (87%)

* Depending on if the patient Case 8 who was found to be infected 1 year after his 2nd-stage revision had an ongoing original infection or a new PJI.

**Table 6 jcm-11-00122-t006:** Functional outcomes in patients undergoing two-stage or multi-staged revisions by time.

	HHS (Median, Range)				Pain (Median, Range)				Activity (Median, Range)			
Surgery/Stage	Pre	12 m	24 m	Last fup	pre	12 m	24 m	Last fup	pre	12 m	24 m	Last fup
Two-stage revision (*n* = 18)	56 (19–92)	74 (12–97)	80 (46–97)	78 (37–88)	30 (10–44)	42 (0–44)	44 (20–44)	42 (10–44)	2 (1–4)	3 (2–5)	3 (2–5)	3 (1–5)
Multi-staged revisions (*n* = 5)	40 (13–73)	76 (60–78	73 (23–76)	77 (73–81)	15 (0–44)	30 (10–44)	42 (10–44)	30 (20–44)	2 (2–2)	2 (2–3)	3 (2–3)	3 (2–4)

Poor < 70, fair 70–79, good 80–89, excellent 90–100, 44 = none, 40 = slight, 30 = mild, 20 = moderate, and 10 = marked; 0 = totally disabled, 1 = bedridden/wheelchair, 2 = sedentary, 3 = semi-sedentary, 4 = light labour, 5 = moderate manual labour, and 6 = heavy manual labour.

**Table 7 jcm-11-00122-t007:** Causes of death.

Cause	Number	Charlson Comorbidity Index Score
Cancer	3	1:7:8
Suicide	1	0
Pericarditis	1	5
Endocarditis	1	6
Pneumonia	1	5
Necrotizing fasciitis	1	1
Renal failure	1	3
Unknown	4	4:5:4:5

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
