# Peer review of "Long-Term Outcomes of Staged Revision Surgery for Chronic Periprosthetic Joint Infection of Total Hip Arthroplasty"

_jcm, 2021, doi:10.3390/jcm11010122_

Round 1

Reviewer 1 Report

Dear Authors,

with great pleasure we have conducted the review of this manuscript for your journal.

In brief, the introduction provides sufficient background, I consider the methods to be appropriate and well described, the results are clearly presented and the conclusion is conclusive and derived from the results.

The manuscript brings insights to treatment approach of periprosthetic infection in PJI. The findings of this study confirm our experience at our own institution. We support the establishment of a scientific foundation for this clinical approach. The content is well presented. The scientific contribution is sound. However, it has to be highlighted that only 7/30 patients underwent mulit-staged revision.

Although this study only includes few patients, the results hold a scientific merit. However, the manuscript has several minor flaws. We would reconsider the manuscript after their revisions.

In the following we address the particular points:

Abstract

Please indicate LoE

Introduction

Introduction provides sufficient information for a not exclusively orthopedic journal. Objectives of this study are clearly stated.

Line 56: I do not consider the scientific evidence for this statement to be that strong: „the articulating design of the interval prosthesis is favored since it may likely benefit the postoperative function while it also maintains limb length and soft tissue tension“

Material and Methods

The methods are very precisely described.

Line 104: minor remark – Can you state how long the mean antibiotic treatment lasted as the discussion over the duration (6 vs 12 weeks) is ongoing? Although not stated as a objective of this study, the antibiotic treatment as a relevant factor in PJI-treatment, especially in multi-staged revisions, should be mentioned.

Line 124: Why did you consider two positive cultures as infection free?

Line 127: We consider patients who were on continuing antibiotic treatment not as infection free per se. Please elaborate.

Results

The Results are clearly presented.

Discussion

The discussion is thoughtful and sound. However, especially in multi-staged revisions, PJI treatment should include a profound antibiotic regimen in consultation with an ID. The antibiotics and the treatment durations have not been debated extensively. Can you add a paragraph to cover the importance of antibiotics and the collaboration with an ID-specialists.

Line 167: in the Results you mentioned two patients who were unfit for reimplantation. Did the spacers remain in the patient? Please discuss this finding.

Line 306-324: very interesting discussion of mortality in PJI. However, the comorbidities of your patients have not been incorporated in this paragraph. I consider the comorbidities as a major contributing factor in the described high mortality. Can you provide this information in the context of the discussion?

Figures

ok

Author Response

Reviewer 1:

In brief, the introduction provides sufficient background, I consider the methods to be appropriate and well described, the results are clearly presented and the conclusion is conclusive and derived from the results.

 The manuscript brings insights to treatment approach of periprosthetic infection in PJI. The findings of this study confirm our experience at our own institution. We support the establishment of a scientific foundation for this clinical approach. The content is well presented. The scientific contribution is sound. However, it has to be highlighted that only 7/30 patients underwent mulit-staged revision.

Although this study only includes few patients, the results hold a scientific merit. However, the manuscript has several minor flaws. We would reconsider the manuscript after their revisions.

Response: We thank the reviewer for these positive comments.

Point 1:  Abstract - Please indicate LoE

Response: We have now indicated the level of evidence to be ‘cohort studies’ on the first page, below the abstract.

Point 2:  Introduction - Introduction provides sufficient information for a not exclusively orthopedic journal. Objectives of this study are clearly stated.

Response: Thank you

Point 3:  Line 56: I do not consider the scientific evidence for this statement to be that strong: „the articulating design of the interval prosthesis is favored since it may likely benefit the postoperative function while it also maintains limb length and soft tissue tension“

Response: We agree with the reviewer’s comment. However, despite the paucity of published evidence for articulating hips functioning better than the excision arthroplasty of the hip this is well recognised in the field. This is why the sentence the reviewer refers to states ‘…it may likely benefit…’.

Point 4:  Material and Methods - The methods are very precisely described.

 Line 104: minor remark – Can you state how long the mean antibiotic treatment lasted as the discussion over the duration (6 vs 12 weeks) is ongoing? Although not stated as a objective of this study, the antibiotic treatment as a relevant factor in PJI-treatment, especially in multi-staged revisions, should be mentioned.

Response: We believe that the mean duration of antibiotic treatment in this cohort is deceiving given the small numbers and skewed results because of the long-term antibiotic given to manage the candida infection in one patient. We have now detailed the length of the antibiotic treatment in the patients in this cohort in a new paragraph labelled ‘Antibiotic treatment’ introduced in the Materials and Methods section on page 4:

“Antibiotic treatment

Antibiotic treatment was prescribed by infectious diseases specialist based on culture results. The interval between planned staged surgeries was 3 months, except for mycotic infections. Antibiotic treatment included 6 weeks of intravenous antibiotic followed by 6 weeks of oral antibiotic. Antibiotics were continued until the second surgery excluding the two patients deemed unfit to undergo the second stage revision. For these patients the oral antibiotics were stopped at 3 months after the first stage revision. For the mycotic infection antibiotic treatment and reimplantation surgery were extended to 2 years. For patients undergoing a second-stage revision, intravenous antibiotics were continued postoperatively until definitive culture results were available 14 days after surgery.”

Point 5:  Line 124: Why did you consider two positive cultures as infection free?

Response: We thank the reviewers for picking up this typo. The text has been edited accordingly on page 4 from “Patients were classified as infection-free if they returned 0-2 out of 5 positive cultures…” to “Patients were classified as infection-free if they returned less than 2 positive cultures with the same organism out of 5 samples with a low virulence organism, and 0 positive cultures out of 5 samples with a high virulence organism (12, 31)… [Parvizi J, et al The 2018 Definition of periprosthetic hip and knee infection: an evidence-based and validated criteria. J Arthroplasty 2018 33(5):1309-1314.e2]. Atkins, B. L., Athanasou, N., Deeks, J. J., Crook, D. W., Simpson, H., Peto, T. E., McLardy-Smith, P., & Berendt, A. R. (1998). Prospective evaluation of criteria for microbiological diagnosis of prosthetic-joint infection at revision arthroplasty. The OSIRIS Collaborative Study Group. Journal of clinical microbiology, 36(10), 2932–2939.]

Point 6:  Line 127: We consider patients who were on continuing antibiotic treatment not as infection free per se. Please elaborate.

Response: We agree with the reviewer that while patients are on antibiotics there continues to be a question mark if the infection has been cured. This is why in the sentence in question by the reviewers we categorise patients that have no clinical or laboratory suggestion of an ongoing PJI and who are still on antibiotic or are less than 1 year since ceasing antibiotic as ‘probably infection free’. There is a clear difficulty in establishing a clear terminology in the field. This is particularly the case for cases in which the current guidelines recommend a lengthy period of antibiotic treatment, and the recidivism of infection is known to be particularly high, i.e. mycotic infections. This is why we called cases in which the infection looked to be cleared but they were still on antibiotic treatment (the mycotic infection was on antibiotic treatment for 2 years before being reimplanted) and within 1 year after finishing antibiotic treatment as “probably infection free” rather than “infection free”.

Point 7: Results - The Results are clearly presented.

Response: Thank you

Point 8: Discussion - The discussion is thoughtful and sound. However, especially in multi-staged revisions, PJI treatment should include a profound antibiotic regimen in consultation with an ID. The antibiotics and the treatment durations have not been debated extensively. Can you add a paragraph to cover the importance of antibiotics and the collaboration with an ID-specialists.

Response: We agree with the Reviewer and at our institution PJI are managed by an MDT attended by orthopaedic surgeons and infectious diseases and microbiology consultants.

However, as this is not an outcome of our study, we felt that the sentence requested by the Reviewer fits better in the Introduction rather than the Discussion. We have now added the following sentence on page 2 in the Introduction, second sentence: “In addition, management of PJI remains controversial (4), failure rates continue to be high (5, 6) outcomes of treatment were shown not to have improved over time (7) and the long-term joint function and quality of life of these patients is low (8). The collaboration between surgeons and infectious diseases and microbiology consultants in the diagnosis and treatment of PJI is seen as essential for improvements in the surgical and medical management of this condition (9).” 

Point 9: Line 167: in the Results you mentioned two patients who were unfit for reimplantation. Did the spacers remain in the patient? Please discuss this finding.

Response: The spacers did indeed remain in the patients.

The sentence on page 6 was edited to reflect this and was changed from “The remaining two patients were medically unfit to undergo second-stage surgery however remained infection-free as per clinical and laboratory findings for >24 months” modified to “The remaining two patients were medically unfit to undergo second-stage surgery however remained infection-free as per clinical and laboratory findings for >24 months with the interval prosthesis remaining in-situ”

Point 10: Line 306-324: very interesting discussion of mortality in PJI. However, the comorbidities of your patients have not been incorporated in this paragraph. I consider the comorbidities as a major contributing factor in the described high mortality. Can you provide this information in the context of the discussion?

Response: A table with the patients’ ASA and Charlson Comorbidity index scores were added to the Materials and Methods section, page 3, of the manuscript. The following sentence was added to the end of the paragraph in question on page 12 in the Discussion section: “Both the comorbidities of these patients as well as the treatments for PJI could have contributed to the high mortality rate observed. Future, larger studies should investigate the independent role of these variables in the mortality of patients with PJI.”

Point 11: Figures - Ok

Response: Thank you.

Reviewer 2 Report

Dear researchers, 

Thank you for the opportunity to review your manuscript entitled " Long-term outcomes of staged revision surgery for chronic periprosthetic joint infection of total hip arthroplasty: a dichotomy of life or death."

The present manuscript is an interesting retrospective analysis of a debated topic such as the staged revision surgery for chronic periprosthetic joint infection of total hip arthroplasty.

It is an interesting and well written article that has some aspects to be checked:

  • I suppose that “a dichotomy of life or death” in the title is excessive. The authors should think to modify the title.
  • Line 147: you should explain the set of criteria for the diagnosis of periprosthetic joint infection.
  • Line 82: the fact that the time between the primary/revision arthroplasty and first-stage revision ranged from 1 to 177 months should be inserted as limitations and be analyzed because it can alter the outcomes and can be linked to various antibiotic treatments.
  • Did you evaluate the comorbidity of these patients or perform the Charlson Comorbidity Index?
  • Why causes of death are unknown for 4 patients? They are due to infectious complications?
  • You should stratify the scores by group (not only first or second stage) because patients underwent different surgical stages (e.g. 5 patients repeated the first-stage).

All the best in your submission!

Author Response

Reviewer 2

Dear researchers, 

Thank you for the opportunity to review your manuscript entitled " Long-term outcomes of staged revision surgery for chronic periprosthetic joint infection of total hip arthroplasty: a dichotomy of life or death." The present manuscript is an interesting retrospective analysis of a debated topic such as the staged revision surgery for chronic periprosthetic joint infection of total hip arthroplasty.

Response: Thank you

Point 1:  It is an interesting and well written article that has some aspects to be checked:

I suppose that “a dichotomy of life or death” in the title is excessive. The authors should think to modify the title.

Response: The Title was changed from “Long-term outcomes of staged revision surgery for chronic periprosthetic joint infection of total hip arthroplasty: a dichotomy of life or death” to “Long-term outcomes of staged revision surgery for chronic periprosthetic joint infection of total hip arthroplasty.”

Point 2: Line 147: you should explain the set of criteria for the diagnosis of periprosthetic joint infection.

Response: We have now included an explanation of the MSIS criteria: “Major, definitive criteria include either the presence of two positive cultures of the same organism or the presence of a sinus tract communicating with the joint or prosthesis. Diagnosis can also be made on a cumulative score of ≥6 points from several minor criteria: elevated serum CRP or D-Dimer (2 points) and ESR (1 point); elevated synovial WBC or LE (1 point); PMN (%) (2 points); CRP (1point); positive alpha-defensin (3 points); positive histology (3 points); purulence (3 points); single positive culture (2 points).”

Point 3: Line 82: the fact that the time between the primary/revision arthroplasty and first-stage revision ranged from 1 to 177 months should be inserted as limitations and be analyzed because it can alter the outcomes and can be linked to various antibiotic treatments.

Response: We do agree with the reviewer that early and late infections can have different outcomes and have added this as a potential limitation of the study. However, none of the longer-term cases (>60 m) had extensive treatment for PJI before they were managed in our department with the only treatment being debridement and short-term antibiotic treatment before being treated by staged revision in our department. None of these cases underwent multi-staged surgeries.

Point 4: Did you evaluate the comorbidity of these patients or perform the Charlson Comorbidity Index?

Response: Thank you for the suggestion. These data have now been added as Table 1 in the Materials and Methods section on page 3.

Point 5: Why causes of death are unknown for 4 patients? They are due to infectious complications?

Response: Cause of death was unobtainable for four patients. However, all patients had multiple comorbidities (CCI ≥4) including acute renal failure, cardiac disease and cancer at their last known admission prior to death. This information has been added into the text on page 10.

Point 6: You should stratify the scores by group (not only first or second stage) because patients underwent different surgical stages (e.g. 5 patients repeated the first-stage).

Response: Response: Patient scores were analysed by patients having either two stage or multi staged revisions. A new table (Table 6) and text on page 10, has been added to the paper.

Reviewer 3 Report

Many appreciations to the authors for having presented a so interesting study about “Long-term outcomes of staged revision surgery for chronic periprosthetic joint infection of total hip arthroplasty: a dichotomy of life or death”

Abstract

The abstract is well structured, and it includes the main results of the study. It underlines the

importance of periprosthetic joint infection as a serious adverse event after hip arthroplasty and

should be better analyzed to increase the post-operative management of patients to reduce the complication rate. Please review and try to express in a different way the second part of the abstract.

Background

The introduction is quite well structured, containing the 3 main aims of the study. However, does

not properly define the relationship between the long-term outcomes of patients who suffer PJI (please try to review better different papers that treat this issue and add it to your paper). Trying to reduce PJI could help prevent also the rising of healthcare costs to treat this kind of patient.

Please add and consider also quoting:

  • Izakovicova P, Borens O, Trampuz A. Periprosthetic joint infection: current concepts and outlook. EFORT Open Rev. 2019 Jul 29;4(7):482-494. doi: 10.1302/2058-5241.4.180092.PMID: 31423332; PMCID: PMC6667982.

  • Fascia and soft tissues innervation in the human hip and their possible role in post-surgical pain. Fede C, Porzionato A, Petrelli L, Fan C, Pirri C, Biz C, De Caro R, Stecco    C. J Orthop Res. 2020 Jul;38(7):1646-1654. doi: 10.1002/jor.24665. Epub 2020 Mar 22.
  • Ting NT, Della Valle CJ. Diagnosis of Periprosthetic Joint Infection-An Algorithm-Based Approach. J Arthroplasty. 2017 Jul;32(7):2047-2050. doi: 10.1016/j.arth.2017.02.070. Epub Mar 2. PMID: 28343826.

Methods

This section contains enough information to understand and possibly repeat the study. In particular, it is well structured in relation to selection patients refers to inclusion and exclusion criteria and the different steps to treat a PJI. However, this section does not reflectand analyze the mean time between one step and others that could be implemented.

This study has two major limitations (provide them in the discussion section): first, the relatively

small sample study: however, this is a single-center study, and data are homogeneous thus eliminating the potential confounding factors. However, a multicentric study with more patients evaluated could better address the aims of the study.

Results

The results presented are quite complete, reflecting the MM section. However, they must be integrated with those coming from MM section.

Discussion

The length and content of the discussion communicate the main information of the paper. Sometimes it is impossible to predict which resistant pathogens it can worsen the prognosis and leading to a lower rate of resolution as illustrated in the paper.

Please discuss two important aspect of this study: the length of antibiotic therapy that could influence the outcome and the timing of different revision surgery at least for those patient reported also quoting others paper who treat similar argument.

Conclusions

The conclusions only reflect and refer to the results of the study, however, others important treatment such as antibiotics therapy appropriate for the specific patients is not evaluated.

References

The references are not up to date. Hence, delate those before 2010 if not essential, replacing them with newer ones and integrate them as suggested.

Tables and Figures

The number and quality of tables are appropriate to transmit the main information of the paper.

However, provide a proper Tables and Figure legends section.

Author Response

Reviewer 3

Many appreciations to the authors for having presented a so interesting study about “Long-term outcomes of staged revision surgery for chronic periprosthetic joint infection of total hip arthroplasty: a dichotomy of life or death”

Response: Thank you

Point 1: Abstract - The abstract is well structured, and it includes the main results of the study. It underlines the importance of periprosthetic joint infection as a serious adverse event after hip arthroplasty and should be better analyzed to increase the post-operative management of patients to reduce the complication rate. Please review and try to express in a different way the second part of the abstract.

Response: Although we don’t disagree with the reviewer that post-operative management can influence complication rates, such analysis was not part of this study. To better reflect this in the abstract, the end of the abstract was edited to read: “The results show that with repeated aggressive surgical treatment, most PJIs of the hip are curable. Ten-years after successful treatment of PJI, functional outcomes and pain are improved and maintained compared to before initial surgery, but this must be balanced against the high 10 year mortality.

Point 2: Background - The introduction is quite well structured, containing the 3 main aims of the study. However, does not properly define the relationship between the long-term outcomes of patients who suffer PJI (please try to review better different papers that treat this issue and add it to your paper). Trying to reduce PJI could help prevent also the rising of healthcare costs to treat this kind of patient.

Please add and consider also quoting:

  • Izakovicova P, Borens O, Trampuz A. Periprosthetic joint infection: current concepts and outlook. EFORT Open Rev. 2019 Jul 29;4(7):482-494. doi: 10.1302/2058-5241.4.180092.PMID: 31423332; PMCID: PMC6667982.
  • Fascia and soft tissues innervation in the human hip and their possible role in post-surgical pain. Fede C, Porzionato A, Petrelli L, Fan C, Pirri C, Biz C, De Caro R, Stecco    C. J Orthop Res. 2020 Jul;38(7):1646-1654. doi: 10.1002/jor.24665. Epub 2020 Mar 22.
  • Ting NT, Della Valle CJ. Diagnosis of Periprosthetic Joint Infection-An Algorithm-Based Approach. J Arthroplasty. 2017 Jul;32(7):2047-2050. doi: 10.1016/j.arth.2017.02.070. Epub Mar 2. PMID: 28343826.

Response: The introduction has been extensively revised to reflect the reviewer’s comments. New literature has been added in the reference list including 2 of the 3 suggested by the reviewer. The paper by Fede et al the reviewer suggests is however irrelevant to the topic discussed.

Point 3: Methods - This section contains enough information to understand and possibly repeat the study. In particular, it is well structured in relation to selection patients refers to inclusion and exclusion criteria and the different steps to treat a PJI. However, this section does not reflect and analyze the mean time between one step and others that could be implemented.

Response: Thank you for these positive comments. We believe we adequately describe the mean time and range of intervals between first and second stage on page 3 of the manuscript. Early and late infections can have different outcomes and have added this as a potential limitation of the study. However, none of the longer term cases (>60 m) had extensive treatment for PJI before they were managed in our department with the only treatment being debridement and short-term antibiotic treatment before being treated by staged revision in our department. None of these cases underwent multi-staged surgeries.

Point 4:  This study has two major limitations (provide them in the discussion section): first, the relatively small sample study: however, this is a single-center study, and data are homogeneous thus eliminating the potential confounding factors. However, a multicentric study with more patients evaluated could better address the aims of the study.

Response: These have been added as limitations of the study in the discussion

Point 5: Results - The results presented are quite complete, reflecting the MM section. However, they must be integrated with those coming from MM section.

Response: We find the 2 sentences in the comment somewhat contradictory. We agree with the Reviewer that our results section is quite complete and reflective of the Materials and Methods section.

Point 6: Discussion The length and content of the discussion communicate the main information of the paper. Sometimes it is impossible to predict which resistant pathogens it can worsen the prognosis and leading to a lower rate of resolution as illustrated in the paper.

Please discuss two important aspect of this study: the length of antibiotic therapy that could influence the outcome and the timing of different revision surgery at least for those patient reported also quoting others paper who treat similar argument.

Response: To clarify the antibiotic treatment, the Materials and Methods section was edited to include a paragraph on page 4 detailing the treatment: “Antibiotic treatment was prescribed by infectious diseases specialists based on culture results. The interval between planned staged surgeries was 3 months, except for mycotic infections. Antibiotic treatment included 6 weeks of intravenous antibiotic followed by 6 weeks of oral antibiotics. Antibiotics were continued until the second surgery excluding the two patients deemed unfit to undergo the second stage revision. For these patients the oral antibiotics were stopped at 3 months after the first stage revision. For the mycotic infection antibiotic treatment and reimplantation surgery were extended to 2 years. For patients undergoing a second-stage revision, intravenous antibiotics were continued post-operatively until definitive culture results were available 14 days after surgery.”

The following sentences were added to page 11 of the Discussion section of the Manuscript: “Several variables might have however influenced our results. Sixteen patients had a DAIR and were commenced on antibiotics prior to having a staged revision. Importantly, in our study all failed first stage revisions were in cases that underwent a prior DAIR. Failed DAIR is thought by some to influence the results of staged revisions for PJI (41, 42). The length of antibiotic treatment is also known to influence outcomes of PJI and importantly the current trend is to reduce the length of intravenous treatment (10, 43).”

Point 7: Conclusions - The conclusions only reflect and refer to the results of the study, however, others important treatment such as antibiotics therapy appropriate for the specific patients is not evaluated.

Response: The conclusions of this study should only refer to our own findings.

Point 8: References - The references are not up to date. Hence, delate those before 2010 if not essential, replacing them with newer ones and integrate them as suggested.

Response: The references have been updated.

Point 9:  Tables and Figures - The number and quality of tables are appropriate to transmit the main information of the paper. However, provide a proper Tables and Figure legends section.

Response: We have formatted the manuscript as per the Journal instructions. All Tables and Figures have legends. To make them more obvious we have edited them to sit just below the respective Tables and Figures.

Round 2

Reviewer 3 Report

The authors answered to my comments properly.

Well done!